# Association between Healthy Dietary Patterns and Self-Reported Sleep Disturbances in Older Men: The ULSAM Study

**DOI:** 10.3390/nu11051029

**Published:** 2019-05-08

**Authors:** Lieve van Egmond, Xiao Tan, Per Sjögren, Tommy Cederholm, Christian Benedict

**Affiliations:** 1Department of Neuroscience, Uppsala University, Sleep Research Laboratory, 751 23 Uppsala, Sweden; xiao.tan@neuro.uu.se (X.T.); christian.benedict@neuro.uu.se (C.B.); 2Department of Public Health and Caring Sciences, Unit of Clinical Nutrition and Metabolism, Uppsala University, 751 22 Uppsala, Sweden; per.sjogren@pubcare.uu.se (P.S.); tommy.cederholm@pubcare.uu.se (T.C.)

**Keywords:** mediterranean diet, healthy diet indicator, sleep problems, elderly population, dietary adherence

## Abstract

To date, little is known about how dietary patterns may link to measures of sleep quality in older subjects, who often suffer from sleep problems. Here, we investigated, in an older male population from Sweden (*n* = 970; aged 71 ± 1 year), whether adherence to the Healthy Diet Indicator (HDI; based on recommendations from the World Health Organization) or the Mediterranean Diet (MD) is linked to sleep disturbances. The diet scores were calculated using a seven-day food diary, and self-reported sleep initiation or maintenance problems were assessed by questionnaires. When adjusted for potential confounders, no associations between dietary scores and sleep parameters were found. In contrast, low consumption of milk and dairy products —one of the dietary features of the MD —was associated with better subjective sleep initiation. This association was, however, not found in men with adequate reports of daily energy intake (~54% of the cohort). To summarize, our findings do not suggest that older men can mitigate perceived difficulties to fall and stay asleep by adhering to either the HDI or MD. Whether low consumption of milk and dairy products can facilitate sleep initiation must be confirmed in future studies by utilizing objective measures of sleep such as polysomnography. Finally, when investigating associations between dietary patterns and sleep, particular attention should be paid to the potential confounder of inadequate reporting of energy intake.

## 1. Introduction

Aging is often accompanied by an increased prevalence of sleep problems. Many older adults complain about more shallow sleep and difficulties with sleep initiation and/or sleep maintenance [1,2]. Although sleep problems in the elderly are primarily driven by natural aging, they may also be a result of poor lifestyle choices. Therefore, it is important to study how lifestyle affects sleep among older subjects. One such sleep-modifying lifestyle factor could be diet [3]. For example, by utilizing a self-reported food frequency questionnaire and actigraphy-measured sleep, it has been shown in a US cohort of 2068 older subjects (mean age 68.6 years; 53% females) that those who reported a moderate–high adherence to the Mediterranean diet (MD) were more likely to sleep longer than subjects who reported a low adherence to the MD. Additionally, they were less likely to report insomnia symptoms when in conjunction with short sleep [4]. Another study including 1639 adults (mean age 72.7 years; 59% females) found that self-reported sleep quality but not sleep duration was positively associated with adherence to the MD [5]. None of the studies found sex differences in the associations between sleep and the MD. This contrasts with findings of a previous study in 2673 men and 3213 women over 64 years of age, where a higher adherence to the MD was linked to a lower prevalence of insomnia symptoms in women but not in men [6]. Given the limited number of studies available and the variability of the results, further research is required before the MD can be recommended as a potential non-pharmacological sleep aid.

With this in mind, the present study utilized reports of sleep initiation and maintenance problems, as well as a seven-day dietary registration, from 970 men aged 71 ± 1 year investigation of the longitudinal Swedish cohort study, Uppsala Longitudinal Study of Adult Men (ULSAM), to examine possible associations with the MD. We also investigated potential associations between the Healthy Diet Indicator (HDI) and sleep. The HDI measures adherence to the nutritional guidelines as defined by the World Health Organization (WHO). Many dietary features of both the MD and HDI have previously been linked to improved sleep patterns. For instance, in a study involving 26 young adults without initial sleep problems, high fiber and low saturated fat and sugar intake were associated with deeper, more restorative sleep [7]. Additionally, MD and HDI are characterized by a high content of plant-based foods, which contain melatonin [8,9]. Melatonin is a hormone produced by the pineal gland. It reduces core body temperature and lowers alertness, thereby increasing sleep propensity [10]. Given the current understanding of diet and sleep, we hypothesized that adherence to either of these healthy dietary patterns would reduce the odds of suffering from sleep initiation and maintenance problems in older men.

## 2. Materials and Methods

### 2.1. Population and Study Design

The Uppsala Longitudinal Study of Adult Men (ULSAM) was initiated in 1970. All 50-year-old men born between 1920 and 1924 and living in Uppsala, Sweden, were invited to participate in a health survey aimed at identifying cardiovascular risk factors (see also http://www.pubcare.uu.se/ULSAM). Although participants attended several consecutive follow-up investigations, assessments relevant for the present study occurred only at the age-70 follow-up investigation. Sleep reports and dietary data were available for 1093 participants (from a total of 1211 men). One-hundred-and-twenty-three participants were excluded due to missing covariate data. Thus, 970 participants were available for the final analysis. All participants gave written informed consent and the study was approved by the Regional Ethical Review Board in Uppsala.

### 2.2. Sleep Variables

All sleep variables were assessed using paper questionnaires. To evaluate sleep initiation problems, participants answered the question: “Do you have difficulties falling asleep at night?”. To screen for sleep maintenance problems, subjects were asked: “Do you often wake up in early hours, unable to get back to sleep?”. Answer options were “yes”, “no”, or “I don’t know” for both questions. The answer option “I don’t know” was treated as not having a sleep problem.

### 2.3. Dietary Assessment

For seven consecutive days, participants recorded their dietary intake in a pre-coded menu-book made and previously used by the National Food Administration (NFA) and Statistics Sweden [11]. Before the assessment, a dietitian gave oral instructions to each participant. Daily intake of energy and nutrients was calculated from the documentation of food and portion size. To ensure reasonable dietary reporting, over- and under-reporters were identified using the Goldberg 2 cut-off for adequate reports of energy intake as modified by Black [12]. This cut-off considers the physical activity level (PAL), basal metabolic rate (BMR; kJ/d), and energy intake (EI; kJ/d). PAL was derived from the “leisure-time physical activity” parameter, which is further described in Section 2.4. BMR was calculated following the equation according to Schofield [13]. EI was calculated from the participants´ food diaries. The estimates of PAL, BMR, and EI can be found in the Appendix A. For each subject, the 95% confidence limit of their EI/BMR was calculated following the Goldberg 2 equation. Utilizing this confidence limit, about half of the included men were classified as adequate energy reporters (*n* = 519, 54%).

To calculate the MD adherence score, the components of the traditional MD were determined from the food diary, including fat quality, vegetables, fruits, cereals, fish, meat, dairy, and alcohol. The median of the adherence to the dietary characteristic in the population was used as a cut-off. Several adjustments in the scoring were made to better tailor the dietary intake to typical Swedish food products and habits. For example, intake of fruit or vegetables and legumes greater than the median of the population was scored with 1. The total MD score ranged from 0 to 8 points, with 8 being highly adherent to the MD. A more detailed description can be found at [14].

The HDI score, that includes both nutrient- and food-based targets, is based on the dietary guidelines from the WHO. The food components included are saturated fatty acids (SFAs), polyunsaturated fatty acids (PUFAs), proteins, total carbohydrates, sucrose, fiber, fruit and vegetables, cholesterol, and fish. Modifications in the nutrient cut-off values were made to better match the Swedish national guidelines. The HDI score ranged from −1 to 8 points, with 8 being highly adherent to the Healthy Diet Indicator. A more detailed description can be found at [15].

### 2.4. Covariates

All covariates were measured at age 70. For a detailed methodological description of the covariates, please find http://www.pubcare.uu.se/ulsam. The exact age at the 70-year follow-up was used as a continuous variable. Waist circumference (continuous variable), weight, height, and blood pressure were measured using conventional methods. BMI (kg/m2; continuous variable) was calculated using weight (kg) divided by height (meters) squared. During one of the measurements at the clinic visit, participants were asked about current smoking behavior (currently smoking vs. non-smoking; treated as a binary variable). Hypertension prevalence was diagnosed if the measured supine diastolic blood pressure was ≥95 mmHg and/or if the participant received anti-hypertensive drug treatment (yes or no; binary variable). Diabetes prevalence was diagnosed via an Oral Glucose Tolerance Test (OGTT). Here, participants ingested 75 g glucose dissolved in 300 mL of water. Plasma glucose and insulin were measured before, and 30, 60, 90, and 120 min after ingestion. Diabetes was diagnosed when the 120 min and one or more of the 30–90 min glucose values were ≥ 11.1 mmol/l (binary variable).

From the self-administrated paper questionnaire, leisure time physical activity (PA) and educational level were used as covariates in this analysis. Leisure time PA was entered as a binary variable into the analysis (non-regular vs. regular PA), with regular PA representing the participant performing an active sport or heavy gardening ≥3 h/week and/or hard physical training or engaging in a competitive sport. Educational level was converted into a binary variable using university education vs. upper secondary school (sixth form) or below. Self-administrated questions about heart infarction (“Have you ever been in hospital because of a heart infarction (clot in the heart)?”), angina pectoris (“Have you ever been told by a doctor that you’ve got angina?”), cancer (“Have you had cancer?”), and joint pain occurrence (“Have you been to a doctor because of any other kind of joint problem?”), were answered with either “Yes”, “no”, or “I don´t know”. To transform these variables into binary variables, the answer option “I don´t know” was treated as “no”. Alcohol intake (continuous variable; as average % of daily energy intake) was derived from the dietary assessment as described in Section 2.3 and entered as a continuous variable into the analysis.

The season of assessment could be a potential confounder when investigating sleep behavior [16]. Thus, we determined the season during which participants came to the test center to undergo clinical investigations. Questionnaires utilized in the present study to assess, e.g., sleep had been filled out shortly beforehand. The season variable was treated as an ordinal variable.

### 2.5. Statistics

Statistical analysis was performed using IBM SPSS Statistics 24 (SPSS Inc. Chicago, IL, USA). Normal distribution of continuous variables was confirmed by approximately symmetrical and bell-shaped histograms. In the first step, possible bivariate associations between independent (i.e., covariates and diet scores) and dependent variables (i.e., sleep initiation and sleep maintenance problems) were analyzed using unpaired Student’s *t*-test and χ^2^ test. As suggested elsewhere [17], independent variables (i.e., covariates and diet scores) with P<0.2 on bivariate tests were then entered into the multivariate logistic regression model (MVLR; enter method). If the 95% confidence interval excluded unity, the test of the statistical hypotheses was considered significant at the 5% level [18].

## 3. Results

### 3.1. Descriptives

Descriptive data of the cohort are summarized in Table 1. Briefly, the total cohort was overweight (BMI: 26.1 kg/m^2^), had a borderline unhealthy waist circumference (according to cut-off values described in [19]), and more than half of the participants reported to be regularly physically active (62.1%). About eleven percent of the participants reported suffering from sleep initiation problems, while eighteen percent recounted sleep maintenance problems. Similar characteristics were found among adequate reporters of daily energy intake. For more information about the cohort, see Table 1.

### 3.2. Association between Adherence to the Mediterranean Diet (MD) and Self-Reported Sleep Problems in Older Men

In the total cohort, a crude analysis showed that adherence to the MD (as measured by the MD score ranging between 0 and 8 points) did not differ between men with sleep initiation problems and those without (mean ± SEM, reports of sleep initiation problems vs. no reports of sleep initiation problems: 3.73 ± 0.14 vs. 3.89 ± 0.05 points, *p* = 0.32, unpaired Student’s *t*-test). Similar results were found among those with adequate reports of daily energy intake (reports of sleep initiation problems vs. no reports of sleep initiation problems: 3.78 ± 0.21 vs. 3.92 ± 0.07 points, *p* = 0.51, unpaired Student’s *t*-test). However, men adhering to a low intake of milk and dairy products—a feature of the MD—had 36% lower odds of reporting sleep initiation problems than those reporting a high intake of milk and dairy products, while adjusting for potential confounders (Table 2). This association was not significant in the group of adequate responders (Table 2). No other associations between dietary features of the MD and sleep initiation problems were found (i.e., *p* ≥ 0.2 for the association with sleep initiation problems on a bivariate test).

In the next step, we investigated possible associations between the MD score and sleep maintenance problems. This analysis revealed that the MD score was not associated with sleep maintenance problems, neither in the full cohort (reports of sleep maintenance problems vs. no reports of sleep maintenance problems: 3.81 ± 0.11 vs. 3.89 ± 0.06 points, *p* = 0.57, unpaired Student’s *t*-test), nor in the group of adequate responders (reports of sleep maintenance problems vs. no reports of sleep maintenance problems: 3.94 ± 0.14 vs. 3.90 ± 0.08 points, *p* = 0.83, unpaired Student’s *t*-test *p* = 0.12). Chi-square testing indicated that the MD components “high intake of cereals” and “moderate alcohol intake” may be potential predictors of sleep maintenance problems, both in the full cohort (*p* = 0.17 and *p* = 0.09, respectively) and among adequate responders (*p* = 0.04 and *p* = 0.126, respectively). However, subsequent multivariate logistic regression analyses did not confirm these assumptions (Table 2).

### 3.3. Association between Healthy Diet Indicator (HDI) and Self-Reported Sleep Problems in Older Men

The HDI score, ranging from −1 to 8, did not differ between men with sleep initiation problems and those without, neither in the full cohort (reports of sleep initiation problems vs. no reports of sleep initiation problems: 3.56 ± 0.20 vs. 3.52 ± 0.06 points, *p* = 0.84, unpaired Student’s *t*-test), nor in the group of adequate responders (reports of sleep initiation problems vs. no reports of sleep initiation problems: 3.53 ± 0.24 vs. 3.37 ± 0.08 points, *p* = 0.53, unpaired Student’s *t*-test). Similar null findings were obtained for the dietary components of the HDI in the full cohort (Table 3). Among men with adequate reports of daily energy intake, the HDI components “low energy intake from saturated fatty acids” and “50–70% of energy from total carbohydrates” appeared to be linked to sleep initiation problems (*p* = 0.08 and *p* = 0.10, as derived from chi-square testing). However, a subsequent multivariate logistic regression analysis, adjusting for potential confounders, did not yield significant associations between these dietary features of the HDI and sleep initiation problems (Table 3).

The HDI score did not differ between men with sleep maintenance problems and those without in the full cohort (reports of sleep maintenance problems vs. no reports of sleep maintenance problems: 3.54 ± 0.13 vs. 3.52 ± 0.06 points, *P* = 0.88, unpaired Student’s *t*-test). Similar results were found in the group of adequate responders (reports of sleep maintenance problems vs. no reports of sleep maintenance problems: 3.45 ± 0.18 vs. 3.38 ± 0.08 points, *p* = 0.73, unpaired Student’s *t*-test). When investigating bivariate associations, the following dietary features of the HDI were identified as potential predictors of sleep maintenance problems in the full cohort: “5–10% of energy from PUFAs”, “10–20% of energy from protein”, “50–70% of energy from total carbohydrates”, and “≥3 g/MJ fiber” (all *p* < 0.2, as derived from chi-square testing). Among those four dietary features, only the variable “50–70% of energy from total carbohydrates” passed the a priori cut-off value of *p* < 0.2 in the group of adequate responders. Importantly, none of the potential dietary predictors of sleep maintenance problems reached significance when adjusted for potential confounders (see Table 3).

## 4. Discussion

The present study, which includes 970 Swedish men aged 71, did not find compelling evidence that adhering to the Mediterranean diet (MD) or Healthy Diet Indicator (HDI) is linked to better subjective sleep initiation and maintenance. However, we found that men reporting a low intake of milk and dairy products, a component of the MD, exhibited lower odds of experiencing difficulty falling asleep, compared with those reporting high consumption of milk and dairy products.

To the best of our knowledge, only three studies to date have investigated the link between adherence to the MD and sleep quality. In line with our findings, a previous study did not observe an association between higher adherence to the MD and insomnia symptoms in men over 65 years old (*n* = 2673). However, in the same study, adherence to the MD was linked to a lower prevalence of self-reported insomnia in older women (*n* = 3213), suggesting possible sex differences in the response to adherence to the MD [6]. In a separate study, using data from 1596 people aged over 59 years, it was further noted that higher adherence to the MD was associated with better subjective sleep quality in both men and women [20]. A more recent study found that consumption of a Mediterranean-style diet was linked to more favorable sleep patterns among older men and women. Participants with moderate–high MD scores were less likely to experience insomnia symptoms together with short sleep duration (which was not assessed in the present study), compared to those with low MD scores [4].

Whereas some attention has been paid to a possible association between the MD and sleep, no study to date has investigated whether adherence to the HDI is linked to improved sleep among older subjects. The HDI incorporates seven WHO recommendations regarding nutrients or food groups. This includes, for example, a high intake of fiber and a low intake of saturated fat, both of which have previously been linked to deeper and more restorative sleep with fewer arousals in an experimental study involving 26 adults (30–45 years; [7]). Contrary to our expectations, in the present study, neither the total HDI score nor the HDI components were associated with sleep initiation or maintenance problems.

One explanation for why no association between overall dietary scores and sleep metrics was observed could be that we did not have information about factors that can potentially modulate the influence of diet on sleep. This includes circadian aspects of food intake, such as the time between the last meal and bedtime. For instance, when overweight individuals with >14 h eating duration ate for only 10–11 h daily for 16 weeks, improved sleep was observed [21]. Additionally, we cannot rule out that the macro- and micro-nutritional composition of the last meal before bedtime, which was not investigated in the present study, may be the most important sleep-modifying feature of dietary patterns. For example, both the HDI and MD are characterized by a high intake of plant-based foods, which can contain the sleep-promoting hormone melatonin [8,9]. In this context, timing the intake of melatonin-containing plant-based foods closer to the intended bedtime may be hypothesized to be more efficacious in promoting sleep, compared to an intake of these foods during other times of the day.

Notwithstanding the lack of association between the diet scores and sleep metrics, we found that low milk and dairy intake was linked to a decrease in sleep initiation problems. Milk is traditionally seen as a tranquilizing and sleep-promoting beverage. Importantly, yet commonly believed, milk alone exerts only minor, if any, effects on sleep [3,22]. In contrast, only when milk is enriched with Horlicks powder (malted milk, as explained in more detail in [3]), tryptophan, or melatonin, positive effects on sleep parameters have been observed [3,22]. One explanation for why we found a negative association between milk and milk product consumption and sleep initiation problems could relate to beta-casein proteins. They naturally represent about 30% of cow´s milk and are suggested to play a role in gastrointestinal problems, such as bloating and abdominal pain [23]. As the discomfort associated with gastrointestinal problems could lead to problems falling asleep [24], this could explain why a lower intake of milk and dairy products was associated with better subjective sleep initiation in the present study.

There are several limitations to this study. Only older men were investigated in this study, making it difficult to generalize our findings to women or other age groups. Therefore, more research is needed to investigate the generalizability of our results. In ULSAM, no objective sleep measures were collected using, for example, actigraphy or polysomnography. Neither was any information gathered about the duration and frequency of insomnia symptoms. Even if no objective sleep data were available, investigating the association between dietary habits and perceived inadequate sleep nonetheless warrants attention as the latter may lead to distress and anxiety [25]. Another limitation of our study is its cross-sectional design. Thus, we cannot rule out inverse causation regarding the association between low milk and dairy product consumption and better sleep initiation. For example, previous studies have shown alterations in food intake after sleep loss, indicating higher total energy and fat intake [26] and increased portion size [27].

Despite these limitations, our study has several strengths. Our analysis was controlled for a variety of potential confounders such as hypertension, diabetes, and smoking. Another strength is that we examined whether misreporting energy intake can affect the association between sleep and diet using the Goldberg criterion [12]. Underreporting and overreporting energy intake both represent major challenges for nutritional epidemiological research [28]. Indeed, almost half the cohort was found to misreport their daily energy intake when using the Goldberg criterion as a cut-off in this study. It is noteworthy that in contrast to what we observed in the full cohort (i.e., including men with both adequate and inadequate reports of energy intake), no association between the variable “low consumption of milk and dairy products” and sleep initiation problems was found in the group of adequate responders. Thus, more studies are needed to examine whether the association between low consumption of milk and dairy products can ease subjective difficulties to fall asleep in older men. It is also worth emphasizing that whenever associations between self-reported dietary patterns and sleep metrics are investigated, particular attention should be paid to the potential confounder of inadequate reporting of energy intake.

## Figures and Tables

**Table 1 nutrients-11-01029-t001:** Cohort characteristics.

Variables	Total Cohort	Adequate Reporters
Number of participants	970	519
Age (years)	71.0 ± 0.6	71.0 ± 0.6
BMI (kg/m^2^)	26.1 ± 3.4	25.1 ± 2.8
Waist circumference (cm)	94.3 ± 9.4	91.7 ± 8.0
Sleep initiating problems (% total)	10.7 ^a^	10.6 ^b^
Sleep maintenance problems (% total)	18.4 ^a^	15.4 ^b^
Diabetes diagnosis (% total)	13.4 ^a^	11.8 ^b^
Hypertension diagnosis (% total)	30.5 ^a^	26.4 ^b^
Heart infarction (% total)	9.6 ^a^	9.1 ^b^
Angina pectoris (% total)	13.5 ^a^	11.8 ^b^
Cancer (% total)	4.8 ^a^	5.6 ^b^
Joint problems (% total)	24.0 ^a^	23.5 ^b^
Currently smoking (% total)	20.0 ^a^	19.5 ^b^
Regular physical activity (≥3 h/week; % total)	62.1 ^a^	60.5 ^b^
Alcohol intake (average % of total daily energy intake)	2.7 ± 3.2	2.4 ± 2.7
University education (% total)	17.4 ^a^	19.3 ^b^
Season of assessment		
Spring (% total)	21.3 ^a^	20.0 ^b^
Summer (% total)	9.6 ^a^	10.2 ^b^
Fall (% total)	42.6 ^a^	44.5 ^b^
Winter (% total)	26.5 ^a^	25.2 ^b^

Values are expressed as mean (± SD) unless otherwise specified. ^a^ percentage of total cohort (*n* = 970). ^b^ percentage of total adequate reporters (*n* = 519). Abbreviations: BMI, Body Mass Index.

**Table 2 nutrients-11-01029-t002:** Association between adherence to the Mediterranean diet and self-reported sleep problems in older Swedish men.

	Sleep Initiation Problems		Sleep Maintenance Problems	
	Full Cohort ^a^ OR [95%CI]	Adeq. Resp. ^b^ OR [95%CI]	Full Cohort ^c^ OR [95%CI]	Adeq. Resp. ^d^ OR [95%CI]
MD score (0 to 8 points)	#	#	#	#
MD subscores (binary)				
High PUFAs/SFAs (> population median)	#	#	#	#
High intake of vegetables and legumes (> population median)	#	#	#	#
High intake of fruit and berries (> population median)	#	#	#	#
High intake of cereals, incl. potato (> population median)	#	#	1.20 [0.85,1.68]	1.63 [0.99,2.68]
High intake of fish (> population median)	#	#	#	#
Low intake of meat and meat products (< population median)	#	#	#	#
Low intake of milk and dairy products (< population median)	0.64 [0.42,0.98]	0.60 [0.33,1.08]	#	#
Moderate alcohol intake ^§^	#	#	0.77 [0.53,1.11]	0.72 [0.41,1.24]

Multivariate logistic regression was utilized to investigate possible associations between adherence to the Mediterranean diet and self-reported sleep problems. Overall, if the 95% confidence interval did not include 1, the adjusted odds ratio was considered significant at the 5% level (shown in bold, [18]). Abbreviations: Adeq. Resp., adequate responders; OR, Odds Ratio; 95%CI, 95% confidence interval; MD, Mediterranean Diet; SFAs, saturated fatty acids; PUFAs, polyunsaturated fatty acids. § Defined as residual adjusted intake of 10–50 g/d and no biochemical signs of alcohol abuse (i.e., aspartate aminotransferase:alanine aminotransferase ratio <2); # Variable was not considered eligible for inclusion into the multivariate logistic regression analysis, as it did not show an association with the sleep variable on a bivariate test (i.e., *p* ≥ 0.2); ^a^ Adjusted for hypertension status, smoking, physical activity, educational status, previous heart infarction, angina pectoris symptoms, and joint problems (all *p* < 0.2 on; bivariate tests); ^b^ Adjusted for waist circumference, BMI, smoking, physical activity, and joint problems (all *p* < 0.2 on bivariate tests); ^c^ Adjusted for waist circumference, BMI, diabetes status, smoking, physical activity, educational status, angina pectoris symptoms, and joint problems. (all *p* < 0.2 on; bivariate tests); ^d^ Adjusted for waist circumference, exact age, and physical activity (all *p* < 0.2 on bivariate tests).

**Table 3 nutrients-11-01029-t003:** Association between adherence to the Healthy Diet Indicator and self-reported sleep problems in older Swedish men.

	Sleep Initiation Problems		Sleep Maintenance Problems	
	Full Cohort ^a^ OR [95%CI]	Adeq. Resp. ^b^ OR [95%CI]	Full Cohort ^c^ OR [95%CI]	Adeq. Resp. ^d^ OR [95%CI]
HDI score (−1 to 8 points)	#	#	#	#
HDI subscores				
0–12% of energy from SFAs	#	1.59 [0.81,3.12]	#	#
5–10% of energy from PUFAs	#	#	0.72 [0.51,1.03]	#
10–20% of energy from protein	#	#	0.82 [0.58,1.16]	#
50–70% of energy from total carbohydrates	#	1.41 [0.71,2.80]	1.28 [0.88,1.87]	1.38 [0.84,2.24]
>10% of energy from sucrose	#	#	#	#
≥3 g/MJ fiber	#	#	1.12 [0.78,1.61]	#
>400 g/day fruit and vegetables	#	#	#	#
0–300 mg/day cholesterol	#	#	#	#
≥35 g/day fish	#	#	#	#

Multivariate logistic regression was used to investigate possible associations between adherence to the Healthy Diet Indicator and self-reported sleep problems. Overall, if the 95% confidence interval did not include 1, the adjusted odds ratio was considered significant at the 5% level [18]. Abbreviations: Adeq. Resp., adequate responders; OR, Odds Ratio; 95% CI, 95% confidence interval; HDI, Healthy Diet Indicator; SFAs, saturated fatty acids; PUFAs, polyunsaturated fatty acids. # Variable was not considered eligible for inclusion into the multivariate logistic regression analysis, as it did not show an association with the sleep variable on a bivariate test (i.e., *p* ≥ 0.2). ^a^ Adjusted for hypertension status, smoking, physical activity, educational status, previous heart infarction, angina pectoris symptoms, and joint problems (all *p* < 0.2 on; bivariate tests); ^b^ Adjusted for waist circumference, BMI, smoking, physical activity, and joint problems (all *p* < 0.2 on bivariate tests); ^c^ Adjusted for waist circumference, BMI, diabetes status, smoking, physical activity, educational status, angina pectoris symptoms, and joint problems. (all *p* < 0.2 on bivariate tests); ^d^ Adjusted for waist circumference, exact age, and physical activity (all *p* < 0.2 on bivariate tests).

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
