# Peer review of "Association between Healthy Dietary Patterns and Self-Reported Sleep Disturbances in Older Men: The ULSAM Study"

_nutrients, 2019, doi:10.3390/nu11051029_

Reviewer 1 Report

The authors try to investigate the link between healthy dietary patterns and sleep disturbances in elderly men. The paper seems interesting but there are a lot of             points that need to be clarified.

The major point that need to be clarified is why they don’t use objective sleep measures, such as actigraphy or polysomnography. All the sleep variables are          assessed using paper questionnaires and the questions seems to be unobjective and not useless to evaluate sleep quality. Moreover no information are available regarding the duration and the frequency of insomnia symptoms.          

About statistics: it's not clear how the ordinal value is being  treated, I feel this should be clarified. It's not clear also if a minimal model has been built via, for example, step-wise selection. In general p-values do not tell the whole story about    the model and in particular in this analysis simply "throwing in"  doesn't seem to extract all the possible information from the data. It's not clear, for example, if there's an effect due to seasonality, as one would have to use a different approach to be able to assess this kind of dependency, rather than simply having the variable in the model.

         Finally, the fact that there is an association between low milk and diary intake and a         decrease in sleep initiation problems seems like a rather bold statement, given that this "relation" seems to disappear in adequate reporters, and the numbers do not fully support this statement.
   Author Response

Dear Reviewer,

Please find below our responses to your comments. We would like to thank you for helping us to strengthen our analyses and further develop our manuscript.

We hope that the revised version is now suitable for publication in Nutrients.

 Sincerely yours,

The authors

Reviewer 2 Report

In elderly sleep difficulties represent an important problem. This study investigate in an elderly man population whether the adherence to healthy diet indicator and the mediterranean diet can influence the sleep. However, the questionnaire is very poor and the diet is not taking in any consideration the influence of phytomelatonin nutrients that may influence the sleep.

Thus, the hypothesis should be supported by a stronger methodological investigation to have stronger data and discussion.

Author Response

Dear Reviewer,

Please find below our responses to your comments. We would like to thank you for helping us to strengthen our analyses and further develop our manuscript.

We hope that the revised version is now suitable for publication in Nutrients.

 Sincerely yours,

The authors

Reviewer 3 Report

General comments: Overall, the paper is well-written and conveys the message nicely. The paper adds to the literature evidence of an association between low dairy consumption and better sleep initiation in older men. There are a few grammar and typo edits to fix throughout the manuscript (e.g., introduction/second line with a space needed after the period). Also, I don’t like the term “elderly” as it tends to be associated with negative stigma, I would prefer the authors use “older” instead.

 Title: I like the title but maybe make it tighter. For example, “Low dairy consumption positively associates with sleep initiation in older men….”

 Abstract: No comments, reads well.

 Introduction: No comments, reads well to me.

 Methods:

-I really like the sensitivity analyses using adequate energy reporters.

-In the methods, the authors note that the distribution of the data were examined but, in the results, there is no mention what came of this examination. Could the authors add a phrase or a sentence to verify what the authors were looking for in these examinations?

-I think the covariate list is adequate but, if possible, the authors may want to examine other disease conditions (e.g., cardiovascular disease [I realize the authors have hypertension but I don’t think that is enough], arthritis, cancer, COPD) as these may influence both diet and sleep.

-Also, do the authors have access to cognitive status (particularly memory recall)? This may play a part in self-reported measures, particularly with diet. Features of sleep may be more consistently recalled though.

 Results:

-I think the results section is well-written.

-It is interesting that the magnitude of the association with low dairy results holds up (not too much difference between 0.60 and 0.63) but the statistical significance doesn’t in adequate energy reporters. The authors should soften this part of the section when saying “the association persisted”.

 Discussion:

-An overall comment is that the authors should replace “gender” with “sex”.

-I think the discussion is well-written.

-In the third paragraph, I would like to see more discussion on authors’ perspectives on why there was no observed association between overall dietary scores and sleep metrics.

-I really like the possible biological explanations for the link between dairy and sleep but need to define “Horlicks” to readers.

-I would like a couple sentences on strengths of the study.

-I prefer the authors to end with a bit more on the future directions with the results from this study but the current sentence is adequate.

 Tables: Nicely formatted and easy to follow the data and results.

Author Response

Dear Reviewer,

Please find below our responses to your comments. We would like to thank you for helping us to strengthen our analyses and further develop our manuscript.

We hope that the revised version is now suitable for publication in Nutrients.

 Sincerely yours,

The authors

Round  2

Reviewer 1 Report

While I still don't think that the questions I posed in my previous review have been fully answered, I feel that the paper is now clear enought for publication.

I would advise authors to look into the language, both as form and as spelling.

Author Response

We would like to thank Reviewer 1 for the positive feedback. We have revised our manuscript after a critical read by a native English speaker. We hope that our manuscript is now suitable for publication.

Reviewer 2 Report

The authors have change the manuscript according with my observation and the manuscript is improved.

Author Response

We would like to thank the reviewer for the positive feedback.

Nutrients EISSN 2072-6643 Published by MDPI AG, Basel, Switzerland RSS E-Mail Table of Contents Alert
Back to Top